# Anthocyanins of Açaí Applied as a Colorimetric Indicator of Milk Spoilage: A Study Using Agar-Agar and Cellulose Acetate as Solid Support to Be Applied in Packaging

**Samiris Côcco Teixeira** [1], **Taila Veloso de Oliveira** [1,*], **Lais Fernanda Batista** [1], **Rafael Resende Assis Silva** [1], **Matheus de Paula Lopes** [2], **Alane Rafaela Costa Ribeiro** [1], **Thaís Caroline Buttow Rigolon** [3], **Paulo César Stringheta** [3] and **Nilda de Fátima Ferreira Soares** [1]

[1] Food Packaging Laboratory, Department of Food Technology, Universidade Federal de Viçosa, Viçosa 36570-900, Brazil
[2] Department of Food Technology, Federal University of Espírito Santo, Alegre 29500-000, Brazil
[3] Laboratory of Pigments and Bioactive Compounds, Federal University of Viçosa, Viçosa 36570-900, Brazil
\* Correspondence: taila.oliveira@ufv.br; Tel.: +55-28-999010521

**Abstract:** Food that is still fit for consumption is wasted in the domestic environment every day, so food packaging technologies are being developed that will monitor the quality of the products in real time. Highly perishable milk is currently one of the products that suffers most from this waste, due to its short shelf life. Active use-by date (AUBD) indicators have been shown to discriminate between fresh and spoiled milk. Colorimetric indicators undergo characteristic changes in their chemical structure, causing abrupt color changes. Among the polymeric materials studied that may function as solid support are cellulose acetate (CA) and agar-agar (AA). The AA colorimetric indicator proved to be more suitable as a solid support due to its ability to maintain the color change properties of the anthocyanin and its high colorimetric performance. The technique was shown to be capable of indicating, in real time, changes in milk quality.

**Keywords:** polymers; shelf-life; food

## 1. Introduction

For effective food distribution, important factors must be taken into consideration, such as the effects of processing, compliance with regulations that aim to ensure safety and quality, and consumer attitudes towards the product. As a consequence of the supply chain not being adapted to these factors, it is estimated that approximately one-third of all food produced for human consumption is lost or wasted [1].

Thus, the technology can be useful in establishing a positive interaction between the consumer and the product through the use of colorimetric indicators in smart packaging, which has gained notoriety due to the simplicity and ease of communication with the consumer [2–4]. Thus, the development of colorimetric indicators from biodegradable polymers has gained prominence. Biodegradable polymers are natural materials that when associated with substances capable of acting as colorimetric indicators, can integrate smart packaging to detect changes due to spoilage in foods [5–7]. Natural dyes such as anthocyanins are sensitive to pH changes and widely used as food freshness indicators [3].

Anthocyanins are non-toxic water-soluble pigments found in fruits and flowers [8]. Regarding the toxicity of anthocyanins, Curtis et al. [9], conducted a parallel, randomized, placebo-controlled trial to examine the effect of chronic anthocyanin consumption on biomarkers of cardiovascular, liver, and kidney function in 52 postmenopausal women. The study concluded that chronic consumption of anthocyanins (500 mg/d) is safe. In addition, several studies prove the importance of consuming anthocyanins to treat diseases [10–13].

Indicators added to anthocyanins can provide rapid qualitative information through visual colorimetric changes caused by the structural change of the pigment. Thus, to indicate the freshness of low shelf-life, nutrient-rich food products and promote food safety and quality through by monitoring them throughout the distribution chain and during storage conditions, active use-by date indicators have been developed [4,14–17].

Active use-by date (AUBD) is a concept that is gaining prominence in research related to the development of food packaging technologies. Active use-by date or colorimetric indicators can discriminate between fresh and spoiled foods, in products such as pasteurized milk. The objective of this research is the development of a new colorimetric indicator for the determination of milk freshness and the investigation of the influence of different polymeric matrices on color changes, using for this purpose, extract of açaí anthocyanins incorporated into cellulose acetate and agar-agar polymeric matrices, to assist in monitoring the quality of this product.

## 2. Materials and Methods

### 2.1. Materials

Cellulose acetate (GS = 2.5; MM = 2,024,000 g·mol$^{-1}$). Acetone was supplied by Rhodia Solvay Group (Santo André, SP, Brazil). Lactic acid solution ($\geq$85%), NaOH and agar-agar were purchased from Sigma-Aldrich (St. Louis, MO, USA). The pasteurized milk was purchased from the local trade in the city of Viçosa, Minas Gerais, Brazil.

### 2.2. Açaí (Euterpe oleraceae Mart.) Fruit Acquisition

Açaí (*Euterpe oleraceae* Mart.) fruits were collected in three cities in Pará State, Brazil, namely: Abaetetuba (latitude: 1°43′42.1′′ S and longitude: 48°52′11.5′′ W), Ilha das Onças latitude: 1°26′34.4′′ S and longitude: 48° 33′10.5′′ W) and Cametá (latitude: 2°14′21.8′′ S and longitude: 49°29′54.3′′ W). The fruits were harvested, selected, washed, and had their seeds removed. The remaining parts (peel and mesocarp) were ground in a blender with the addition of distilled water for 3 min at a ratio of 3:1 fruit/water. The resulting homogeneous pulp was frozen in a deep freezer at −60 °C + 2 °C until the moment of analysis and used as raw material for the whole experiment.

### 2.3. The Acquisition of the Crude Phenolic Extract

Approximately 50 g of açaí was added to a solution containing ethanol/water (75/25%), then acidified with hydrochloric acid to pH 2.0 ± 0.1. The ultrasound-assisted extraction was performed in an ultrasonic bath (UltraclEAner 1400A, Unique, SP, Brazil) operating at 40 kHz, 40 °C ± 2 °C, for 50 min. The resulting extracts were vacuum filtered using Whatman no. 1 filter and quantitatively transferred to a volumizing flask. Subsequently, the extracts were concentrated under vacuum using a rotary evaporator (IKA RV 10 digital) at a maximum temperature of 50 °C to eliminate alcohol until approximately 7° Brix of total soluble solids (TSS) content. The soluble solids content (° Brix) was determined by direct reading in a Leica AR 200 refractometer (New York, NY, USA). This crude phenolic extract was stored in an amber flask under freezing in the ultrafreezer (approximately −60 °C) until its use for application on CA and AA films [18].

### 2.4. Determination of Total Anthocyanins

The determination of total anthocyanins in açaí extracts (AE) was performed by the single pH method described by Fuleki and Francis [19]. The absorbance of the diluted solution of the açaí extract was measured with a UV-VIS spectrophotometer (Shimadzu 1601Pc, Kyoto, Japan) at a wavelength of 535 nm. The results were expressed as mg of anthocyanins per 100 g of açaí pulp.

*2.5. Film Preparation*

2.5.1. Cellulose Acetate Films

A polymeric dispersion of CA in acetone at a ratio of 1:10 (*w/v*) was prepared. The dispersion was sealed and kept at rest for 24 h. After the elapsed time, 5 mL of anthocyanin extract was added, followed by manual homogenization for 2 min. The final polymeric dispersion was poured into 100 mm Petri dishes and solvent evaporation was performed at room temperature (25 °C + 2 °C). The films were stored under vacuum in polyethylene/nylon packaging until the time of use.

2.5.2. Agar-Agar Films

A 4% by weight agar-agar solution was dissolved in deionized water under constant stirring at 60 °C + 2 °C and mixed in 5 mg/L anthocyanin extract. Then, the dispersion was poured into a 100-mm-diameter Petri dish, where it remained for solvent evaporation at 25 °C + 2 °C. The anthocyanin film was removed from the petri dish and cut into 10 mm-diameter-circles. The films were stored under vacuum in polyethylene/nylon packaging until use.

*2.6. Scanning Electron Microscopy (SEM)*

Micrographs of the films were obtained using a scanning electron microscope (model TM3000, Hitachi Hi-Tech, Tokyo, Japan). Samples of each film, with dimensions of $(0.2 \times 0.5 \text{ cm}^2)$, were fixed on stubs with the aid of tweezers on a conductive double-sided carbon tape (20 to 30 nm). The electron-accelerating voltage was used in automatic mode. The magnification of the images obtained was $800\times$.

*2.7. Infrared Spectroscopy (FT-IR)*

The infrared spectra of the films were obtained with the aid of FT-IR equipment NICOLET 6700 (Thermo Scientific, Waltham, MA, USA), with 32 scans and a resolution of $4 \text{ cm}^{-1}$, operating in the range 4000–700 $\text{cm}^{-1}$.

*2.8. Colorimetric Response of the Extract to Lactic Acid*

The chromatic characteristics of the 10 mL of anthocyanin extract (204.3 ± 6.70 mg of anthocyanins per 100 g of açaí pulp respectively), adjusted to pH 7.0, 6.0, 5.0, 4.0, 3.0 and 2.0 (using lactic acid to adjust the pH's), were measured using a colorimeter Colorquest® XE colorimeter (HunterLab, Reston, VA, USA).

*2.9. Colorimetric Response of the Film to Lactic Acid*

The chromatic characteristics of the anthocyanin-incorporated films were analyzed upon insertion into lactic acid solution adjusted to pH 4.0, 5.0, 5.5, 6.0, and 6.8 by adding NaOH and lactic acid. After 15 min of exposure under the above conditions, the materials were removed and the coordinates of the films of AC and AA were determined using a colorimeter Colorquest® XE colorimeter (HunterLab, Reston, VA, USA). The resulting data were recorded as the parameters L*, a*, b*, and total color difference (ΔE). ΔE was calculated in terms of Equation (1).

$$\Delta E^* = \sqrt{\left[(\Delta L^*)^2 + (\Delta a^*)^2 + (\Delta b^*)^2\right]} \tag{1}$$

where L*, a*, and b* are the color parameters of lightness, red–green chromaticity index, and yellow–blue chromaticity index after sensing, respectively. ΔE color values of milk at pH 6.8 or sample acidified to pH 6.8 were used as the reference and a higher ΔE denotes greater color change compared to the reference material.

*2.10. Colorimetric Response of Milk Samples to Lactic Acid*

The ability of anthocyanin films to discriminate between fresh and spoiled milk by a color change was analyzed under the presence of lactic acid in milk at pH 6.8, 6.0, 5.5, 5.0,

and 4.5. The anthocyanin films were then submerged in 5 mL of the pH-adjusted milk. After 15 min, the materials were removed and photographed for digital colorimetric analysis.

Images of the colorimetric indicators were captured using a smartphone (Apple iPhone). Images were captured at 30 cm under vertical illumination with a white background in 3 repetitions. These images were used to extract colorimetric data from 15 different points, using the software Digital Colorimeter © Microsoft, to obtain the CIELab coordinates or L*, a*, b* values. These data were used to quantify the color of the intelligent indicators.

### 2.11. Acidity and pH Analysis

Milk acidity was measured by the titration method, in which 10 mL of milk and 20 mL of sterile purified water were titrated with 0.1 mol/L NaOH at 25 °C + 2 °C, using alcoholic phenolphthalein as a color indicator. The pHs were determined by adding 100 mL of water to 10 mL of milk, natural or pH-adjusted, and stirred to suspend milk particles. The tests were performed in triplicate.

### 2.12. Statistical Analysis

The color data of the indicators were submitted to ANOVA and Tukey's test (with a significance level of 5% probability). All statistical analyses were performed with the statistical program Minitab version 17 and OriginPro 8.5.

## 3. Results and Discussion

### 3.1. Chromatic Changes for the Açaí Anthocyanins Extract Solution

The açaí extract was obtained from three different municipalities in Pará state, Brazil. The values of total anthocyanins determined were $76.13 \pm 14.77$ mg of anthocyanins per 100 g of açaí pulp for Abaetetuba; $46.88 \pm 25.98$ mg of anthocyanins per 100 g of açaí pulp for Ilha das Onças and $204.3 \pm 6.70$ mg of anthocyanins per 100 g of açaí pulp for Cametá. Thus, for this study, we used the third county extract ($204.3 \pm 6.70$ mg of anthocyanins per 100 g of açaí pulp), in which the concentration of anthocyanins determined was higher than those, from different fruit sources reported by several authors, such as $53.74 \pm 1.75$ mg/100 g in purple cabbage [20] and 106 mg/100 g in blackberry [21].

According to Weston et al. [20], the chromatic transition of anthocyanins is usually monitored by spectroscopic techniques, but it is limited to transparent solutions, and therefore this is not effective for various materials. Therefore, for visual detection and quantification of the color change of anthocyanins, it may be possible to use the color parameters L*, a* and b*.

As shown in Figure 1, the AE solution pH 2.0 exhibited a red color. As the pH increases, the color of the solution changed to pink (pH 2.0), light yellow (pH 5.0), brown (pH 6.0), and dark blue (pH 7.0). The color changes can be attributed to the structural changes of the anthocyanins present in açaí, which are cyanidin 3-O-glucoside and cyanidin 3-O-rutinoside [22]. The coordinates a* and b* decrease with increasing pH, indicating color change from red (a*) and yellow (b*), to dark tones (Table 1). Thus, the anthocyanin extract can be incorporated into solid matrices to indicate milk freshness, since its colorimetric changes were evident.

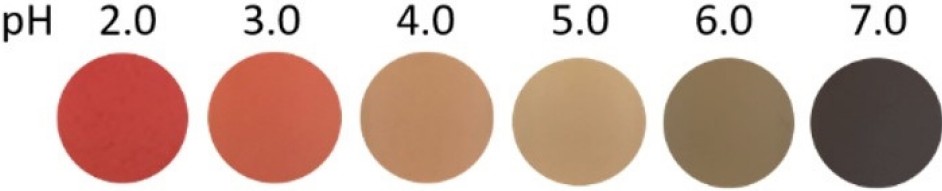

**Figure 1.** Photographs of the anthocyanin solution showing the colorimetric response to lactic acid at different pH levels.

**Table 1.** Total color change values for the AE.

| pH | L* | a* | b* |
|---|---|---|---|
| 2.0 | 46.16 ± 0.42 ᵃ | 49.91 ± 0.40 ᵃ | 31.28 ± 0.10 ᵃ |
| 3.0 | 52.80 ± 0.10 ᵇ | 39.81± 0.53 ᵇ | 32.59 ± 0.06 ᵇ |
| 4.0 | 61.38 ± 0.26 ᶜ | 19.10 ± 0.32 ᶜ | 26.70 ± 0.48 ᶜ |
| 5.0 | 64.66 ± 0.79 ᵈ | 14.38 ± 0.19 ᵈ | 7.49 ± 0.28 ᵈ |
| 6.0 | 49.35 ± 0.16 ᵉ | 4.07 ± 0.08 ᵉ | 19.73 ± 0.42 ᵉ |
| 7.0 | 26.94 ± 0.26 ᶠ | 2.93 ± 0.02 ᶠ | 2.97 ± 0.27 ᶠ |

ᵃ⁻ᶠ Different superscripts in the same parameters indicate significant differences ($p < 0.05$).

To achieve the ultimate goal of the study, the development of an active wearable date using the different properties of CA and AA was carried out to choose the optimal polymeric matrix that provides a visual colorimetric change of the sensor due to milk pH modification.

### 3.2. Color Change on Cellulose Acetate (CA)

The color change of the CA indicator at different pH values is shown in Figure 2, and Table 2 shows the colorimetric parameters. Cellulose acetate is a biodegradable polymer that comes from the acetylation of cellulose, the most abundant natural polymer; its use as a material for making films has been widely reported [4,23–25].

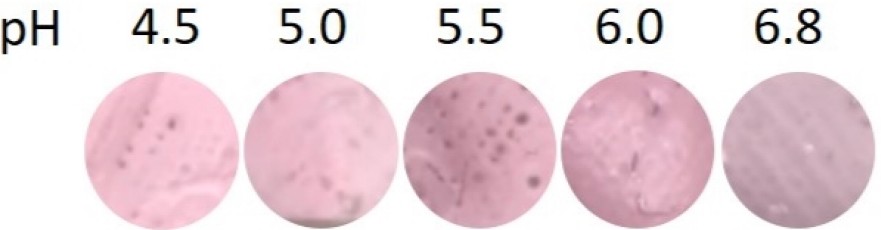

**Figure 2.** Photographs of the AC sensors with anthocyanins showing the colorimetric response to lactic acid at different pH levels.

**Table 2.** Total color change values for the AC colorimetric indicators.

| Sample | pH | L* | a* | b* | ΔE |
|---|---|---|---|---|---|
| | 6.8 | 74.81± 0.08 ᵃ | 16.37 ± 0.42 ᶜ | −1.77 ± 0.47 ᵇ | 0.00 ± 0.00 ᵇ |
| | 6.0 | 77.89 ± 0.54 ᵃ | 17.07 ± 0.06 ᶜ | −2.05 ± 0.41 ᵇ | 8.47 ± 0.42 ᵃ |
| AC | 5.5 | 75.09 ± 0.81 ᵃ | 21.01 ± 0.15 ᵃ | −0.58 ± 0.20 ᵃ | 9.37 ± 1.63 ᵃ |
| | 5.0 | 76.73 ± 1.16 ᵃ | 19.1 ± 0.46 ᵇ | −0.56 ± 0.33 ᵃ | 10.14 ± 1.01 ᵃ |
| | 4.5 | 75.32 ± 4.13 ᵃ | 20.55 ± 0.36 ᵃ | −0.71 ± 0.05 ᵃ | 10.99 ± 1.27 ᵃ |

ᵃ⁻ᶜ Different superscripts in the same parameters indicate significant differences ($p < 0.05$).

The L* value remained the same for all pH values; the a* value increased when the pH value decreased; and the b* value for pH 4.5, 5.50 and 5.5 remained the same. These results indicate that the colorimetric changes that occurred in the indicator with CA were not efficient for use as a quality indicator for milk. Furthermore, it is evident from the ΔE values that there was no statistical difference; therefore, given the color pattern of pH 6.8, the colors of the other pH's are not distinguishable to the human eye.

The slight color change of the CA sensors can be explained by analysis of the SEM images. The microscopic images show that there was a possible immobilization of the anthocyanin molecules in the CA polymer matrix. Therefore, they show no potential for color change. This may have been triggered by the hydrolysis reaction of the CA molecules, which leads to the release of acetyl groups, which may make the CA molecule reactive with the existing anthocyanin molecule. Several studies have reported that hydrolysis of cellulose acetate can occur under different pH and temperature conditions. The rate

of hydrolysis can occur at different temperatures, including 23 °C, and at different pH values [26]. Due to the immobilization of the anthocyanin compounds in the polymer matrix, there was no interaction with the water contained in the lactic acid solution, since CA has a hydrophobic character compared to AA. Since the color change mechanism depends on the pH-controlled ionization of the anthocyanin molecule, the system needs to be hydrated [20,27,28].

### 3.3. Color Change on Agar-Agar (AA)

Natural polymers, such as agar-agar, are biodegradable due to their chemical composition being predominantly composed of carbon and oxygen, unlike petroleum-derived polymers, which have predominantly carbon-carbon bonds [29]. Agar-agar is a polysaccharide that constitutes the main structural component in the cell walls of algal species belonging to the class Rhodophyceae (red algae). It is composed mainly of agarose and agaropectin. The gelling fraction, AA, consists of repeating units of alternating groups of β-D-galactopyranose and 3,6-anhydro-α-l-galactopyranose. Agaropectin has a similar structure to agarose, but contains 5–10% sulfated esters and other residues, such as methoxyl groups and pyruvic acid.

Figure 3 shows the colorimetric transition of AA films with anthocyanin extract when exposed to different pH values. Table 3 shows that the L* values decreased until pH 5.5 and at pH 5.0 and 4.5 showed a gradual increase, indicating that the luminosity was significantly different among the pH values close to the deterioration value of the milk. For pH 6.8, the a* coordinate showed a low value with a propensity to transition from green to red and the b* value was also considerably lower compared to the other treatments, showing a slightly blue/gray color, indicating that it is possible to notice the colorimetric difference between the other pH values. In addition, the ΔE values were significantly different and showed gradual increases with decreasing pH, indicating greater visual perception of color transition.



**Figure 3.** Photographs of the AA sensors with anthocyanins showing the colorimetric response to lactic acid at different pH levels.

**Table 3.** Total color change values for the AA colorimetric indicators.

| Sample | pH | L* | a* | b* | ΔE |
|--------|-----|-----|-----|-----|-----|
| AA | 6.8 | 40.40 ± 2.25 [a] | 0.16 ± 0.27 [d] | −0.04 ± 0.04 [c] | 0.00 ± 0.00 [d] |
| | 6.0 | 34.78 ± 0.76 [b] | 9.33 ± 0.32 [c] | 2.72 ± 0.43 [b] | 11.47 ± 1.20 [c] |
| | 5.5 | 34.91 ± 0.80 [b] | 9.17 ± 0.28 [c] | 2.74 ± 0.41 [b] | 11.54 ± 1.03 [c] |
| | 5.0 | 36.36 ± 1.03 [b] | 11.64 ± 0.37 [b] | 6.64 ± 0.25 [a] | 13.56 ± 1.00 [b] |
| | 4.5 | 38.88 ± 0.44 [a] | 9 ± 0.39 [a] | 7.28 ± 0.37 [a] | 16.31 ± 0.89 [a] |

[a–d] Different superscripts in the same parameters indicate significant differences ($p < 0.05$).

Furthermore, the chromatic properties of the AA sensor determined in this study can distinguish fresh milk from spoiled milk through the color transition from blue/gray to red. This fact can be attributed to the increased hydration caused by the polymer matrix, which is composed of a polysaccharide, as it retains a large amount of water, thus preserving the non-ionized species of anthocyanin molecules that have the potential to change color [20].

### 3.4. Scanning Electron Microscopy (SEM)

Figures 4 and 5 present the SEM images of the fractured surfaces and cross sections of the AC and AA sensors. Notably, it is possible to observe that both the agar-agar (Figure 5) and cellulose acetate films (Figure 4) have agglomerates that may be associated with not fully dispersed compounds, stemming from the production method. On the contrary, the presence of particles that are attributed to anthocyanins in CA films (Figure 4b) which have a semi-rounded shape without the formation of agglomerates, is evident and occurs by the low interaction between the components of anthocyanins, mostly hydrophilic, and CA having high hydrophobicity [30]. These particles are not observed in Figure 4a, confirming the presence of anthocyanins in the CA colorimetric indicator, in agreement with the FT-IR and TGA analysis.

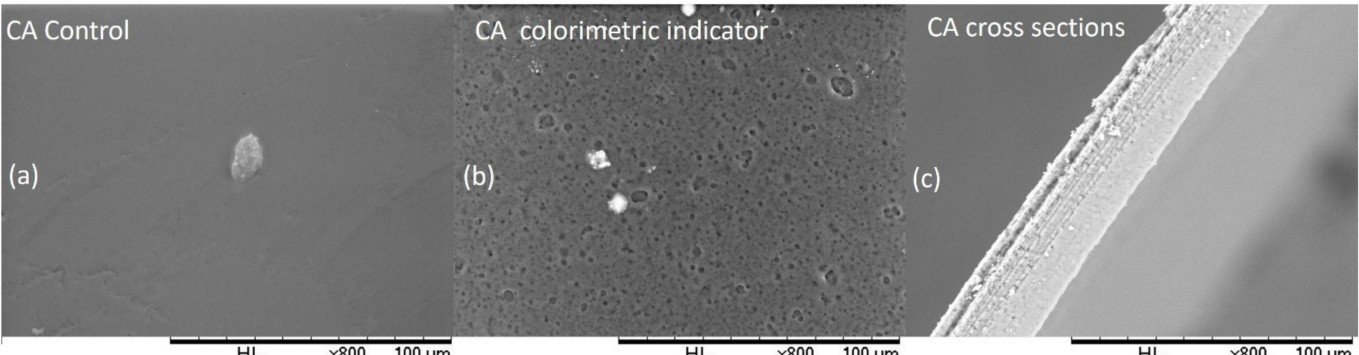

**Figure 4.** SEM of the AC sensors. (**a**) Film control, (**b**) CA indicator colorimetric, and (**c**) cross sections. The left micrograph shows the surface of the sensors and the right micrograph illustrates the film cross section. All magnifications are 800×.

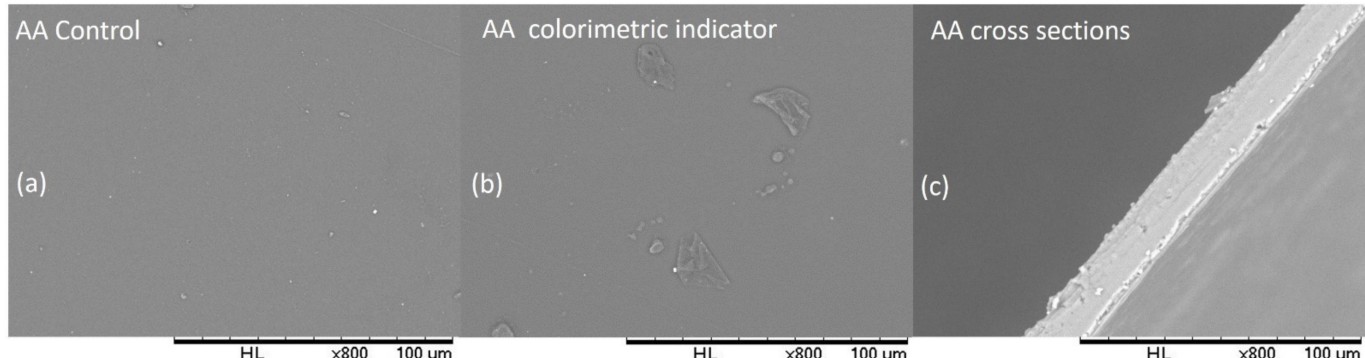

**Figure 5.** SEM of the AA sensors. (**a**) Film control, (**b**) AA indicator colorimetric and (**c**) cross sections. The left micrograph shows the surface of the sensors and the right micrograph illustrates the film cross section. All magnifications are 800×.

On the contrary, in AA films it was not possible to detect the presence of anthocyanins (Figure 5a,b), reported previously in AC films, suggesting the successful incorporation of the extract, which may be due to the interactions between the phenolic hydroxyl groups of anthocyanins with the hydroxyl groups of the agar-agar, forming a semi-continuous matrix; this behavior was also reported in the development of active films of fish gelatin, developed by Uranga et al. [14]. The cross-sectional surface of the films shows the presence of particulates more evidently in the AC films, and the AA film shows a smooth structure with small insoluble particles and the absence of bubbles or grooves.

### 3.5. Infrared Spectroscopy (FT-IR)

FT-IR spectra of AA and AC sensors (with and without EA) were analyzed to verify the chemical interactions between the polymer chains and the extract (Figure 6). In the AC and AC/EA spectra, characteristic peaks of cellulose acetate were observed at 1745 cm$^{-1}$, 1639 cm$^{-1}$, 1430 cm$^{-1}$, and 1371 cm$^{-1}$, which correspond to the stretching of the carbonyl ester, C=C strain, asymmetric angular CH$_3$ strain, and symmetric angular CH3 strain, respectively [31–33]. Peaks at 1234 cm$^{-1}$ and 1049 cm$^{-1}$ represent the C-O stretching of the acetyl group and the symmetric primary alcohol stretching of the AC molecule [34,35]. When the EA was added into the polymeric matrix to produce the film, an appearance of a long band at 3409 cm$^{-1}$ can be observed, which was attributed to the stretching vibration of the hydroxyl group of alcohols and phenols, confirming the presence of EA [36].

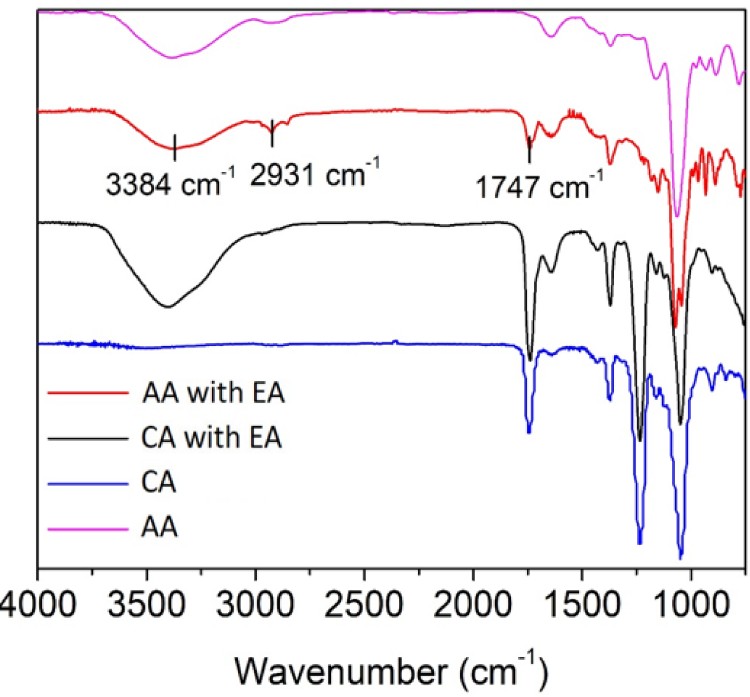

**Figure 6.** FT-IR spectra of AA and CA sensors, with and without anthocyanins.

In the spectra of AA/EA films, a broadband can be observed at 3384 cm$^{-1}$ and is attributed to the stretching vibrations of free hydroxyl groups besides the hydroxyl groups involved in inter- and intra-molecular linking. The bands observed at 2931 cm$^{-1}$ and 2856 cm$^{-1}$ were attributed to the methoxy groups in the structure of the AA molecule [31,37]. At 1747 cm$^{-1}$ a characteristic stretching vibration band of the carbonyl ester group was observed, present in the açaí anthocyanin extract molecules, which is not found in the spectrum of pure AA, confirming the presence of the compound. At 1637 cm$^{-1}$ there was the presence of a peak caused by the elongation of the peptide conjugate bond formed by amine (NH) and acetone (CO) groups [38]. The peak at 1373 cm$^{-1}$ corresponds to the sulfate ester. The peak 1068, 1041 and 935 cm$^{-1}$ corresponds to the C=O stretching vibration of 3,6-anhydrogalactose [39,40].

### 3.6. Thermogravimetric Analysis (TGA)

The thermal stability of AA and CA films, with and without anthocyanin extract, was investigated using the thermogravimetric analysis (TGA) and thermogravimetric derivatives (DTG), and the results were expressed in terms of mass loss (%) as a function of temperature (°C) ranging from 25 °C to 600 °C, as can be observed in Figure 7. The DTG curve indicates that the thermal degradation of the CA films, without anthocyanin extract addition, occurred in two distinct stages. The initial phase occurred between 25 °C

and 120 °C temperatures, and the mass loss was related to the free water evaporation. The second event happened between the 312.8 °C and 394.1 °C temperatures, with mass loss of 74% (m/m), which occurred due to the decomposition of the CA polymer [23,41]. For the CA polymer, with anthocyanin extract addition, three main events could be observed. The first step occurred between, 21.93 °C and 78.70 °C, with approximately 25% (m/m) of mass loss, and was attributed to the free water evaporation and low-molecular-weight compound loss and loss of volatile compounds from anthocyanin. The second stage occurred between 280.14 °C and 380.20 °C, with 57% (m/m) of mass loss attributed to polymer decomposition. The third stage occurred between 380.20 °C to 534.29 °C and were assigned to degradation and oxidation reactions of organic materials [36,42]. It can be seen, however, that the incorporation of AE reduced the stability of the sensors but was not sufficient to impair the processing of the sensors.

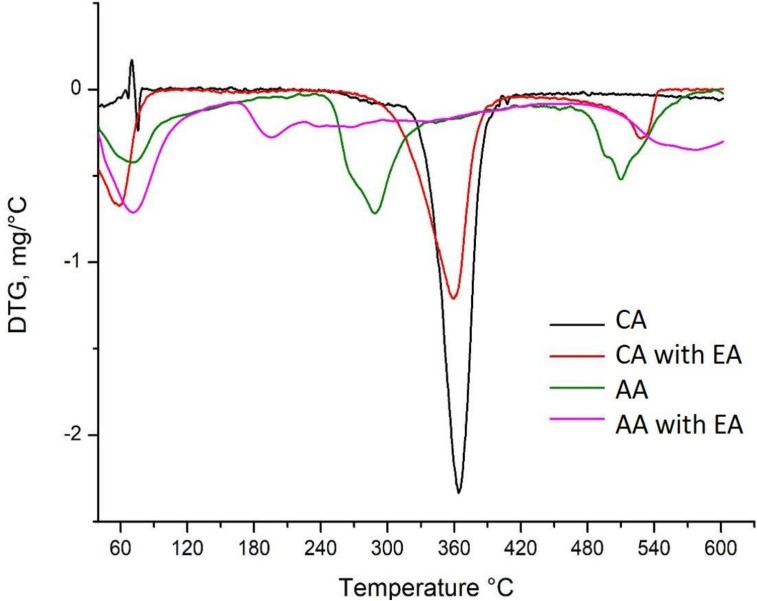

**Figure 7.** Thermogravimetric derivative (DTG) of AA and CA films, with and without anthocyanin extract incorporation.

The DTG curves of the AA films, with and without anthocyanin extract addition, indicated that the thermal degradation occurred in three distinct steps. The first step occurred in a range of 25 °C to ~100 °C due to free water evaporation, with mass loss of 29% for AA films without anthocyanin extract addition and 44% for AA films incorporated with anthocyanin extract. The difference in mass loss can be attributed to the hydrophilic compounds present in the anthocyanin extract. The second stage occurred in a range of 170 °C to 230 °C for AA agar films incorporated with anthocyanin extracts, and the mass loss was 52%, while for AA films without anthocyanin extract addition, 40% mass loss occurred at temperature range 245 °C to 325 °C, which was the main thermal degradation process. This event was induced by the removal of organic functional groups from the agar-agar matrix and decomposition of the extract [43]. The third step occurred at temperatures above 508 °C and can be attributed to the carbonization of organic materials. It can be observed that the AA films with added anthocyanin extract decreased the thermal stability, indicating a strong physical interaction with the polymer components; although the decrease in thermal stability is relevant, the degradation will not cause harm in the application of the sensor because its use occurs at mild temperatures of around 7 °C to 25 °C.

### 3.7. Detection of Spoilage in Milk

Normative Instruction no. 76, of 26 November 2018, stipulates that raw refrigerated milk must present acidity values between 0.14 and 0.18 g of lactic acid/100 mL in milk.

To validate the sensors as a function of change in acidity, the methodology of the Manual of Official Methods of Analysis of Foods of Animal Origin was used and the results are expressed in Table 4 [44]. Fresh milk showed acidity of 0.174 g/100 g of lactic acid, which increased to 0.3075 g/100 g of lactic acid, suggesting deterioration of the milk. Higher concentrations of lactic acid completely spoiled the milk, leading to pH change.

**Table 4.** Milk pH and acidity values.

| pH | Grams of Lactic Acid/100 mL |
|---|---|
| 4.5 | 0.69768 |
| 5 | 0.54162 |
| 5.5 | 0.3672 |
| 6 | 0.30753 |
| 6.8 | 0.17442 |

The colorimetric indicators made with cellulose acetate did not change statistically for all color parameters (L*, a* and b*), remaining pink regardless of pH values (Figure 8A and Table 5). The colorimetric indicators made with agar-agar presented the greatest color change, from greenish blue to pink, when the pH of the milk decreased from 6.8 to 6.0. These results can be observed by the coordinates a* and b*, which increased with the reduction in pH. Therefore, the AA indicators were adequate to detect variations in the pH of the milk (Figure 8B and Table 5).

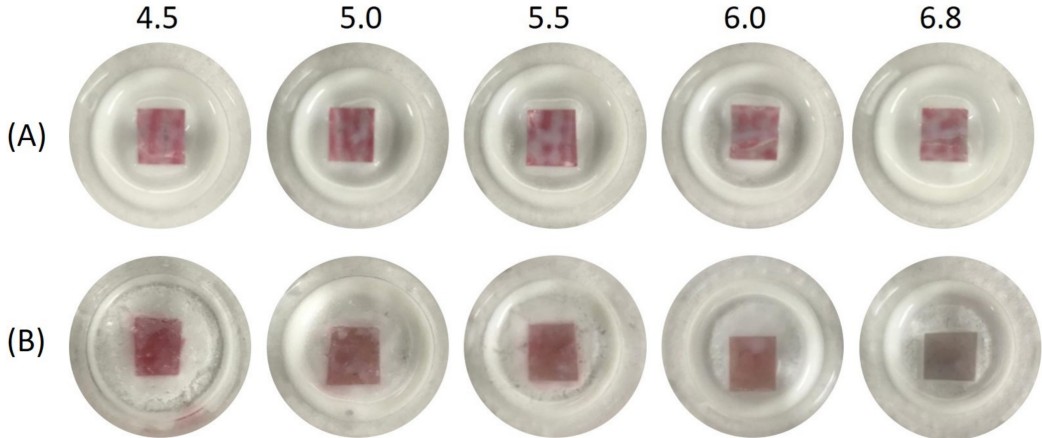

**Figure 8.** The color behavior of the CA (**A**) and AA (**B**) sensors under milk acidification by the addition of lactic acid (excess milk was removed for better visualization of color changes when capturing the photo).

**Table 5.** Total color change values for the AC and AA colorimetric indicators.

| Sample | pH | L* | a* | b* | ΔE |
|---|---|---|---|---|---|
| AC | 6.8 | 46.96 ± 1.17 [a] | 20.80 ± 1.06 [a] | 5.86 ± 0.26 [a] | 0.00 ± 0.00 [b] |
| | 6.0 | 46.68 ± 0.84 [a] | 21.50 ± 0.46 [a] | 5.62 ± 0.14 [a] | 5.74 ± 0.60 [a] |
| | 5.5 | 46.43 ± 0.70 [a] | 20.92 ± 0.95 [a] | 5.36 ± 0.24 [a] | 6.63 ± 0.84 [a] |
| | 5.0 | 46.28 ± 0.77 [a] | 23.29 ± 0.61 [a] | 5.67 ± 0.24 [a] | 7.22 ± 0.84 [a] |
| | 4.5 | 44.75 ± 0.69 [a] | 23.4 ± 0.89 [a] | 5.61 ± 0.2 [a] | 6.83 ± 0.6 [a] |
| AA | 6.8 | 42.37 ± 0.65 [ab] | 3.96 ± 0.26 [c] | 6.53 ± 0.33 [b] | 0.00 ± 0.00 [d] |
| | 6.0 | 42.91 ± 1.22 [a] | 15.13 ± 0.61 [b] | 9.22 ± 0.10 [a] | 11.36 ± 0.38 [c] |
| | 5.5 | 39.94 ± 0.78 [bc] | 16.82 ± 0.24 [ba] | 9.29 ± 0.19 [a] | 13.47 ± 0.25 [b] |
| | 5.0 | 40.57 ± 0.41 [abc] | 15.50 ± 0.33 [ba] | 9.52 ± 0.22 [a] | 11.71 ± 0.37 [c] |
| | 4.5 | 38.35 ± 1.73 [c] | 17.67 ± 0.74 [a] | 9.43 ± 0.30 [a] | 20.55 ± 0.37 [a] |

[a–d] Different superscripts in the same parameters indicate significant differences ($p < 0.05$).

## 4. Conclusions

This study provides a comparative evaluation of two colorimetric indicators, incorporating anthocyanin extract into agar-agar or acetate cellulose solid base, which can be used to detect the freshness of numerous foods. The incorporation step into polymeric matrices was proved by SEM and FT-IR analyses, and thermogravimetric analysis was evaluated to certify the thermostability of the sensors, which decreased with anthocyanin extract addition. However, the thermal behavior did not impair the color change and its application in food packaging. The developed agar-agar sensor can be used to evaluate the freshness of milk since there was a synchronicity between the color change and pH increase. Of the materials examined, a sensor with agar-agar and anthocyanins is cited as the ideal material with the best colorimetric performance, due to its hydrophilic character. The development of AUBD will reduce food waste by decreasing the proliferation of greenhouse gasses. Further work will be conducted to make the sensors more sensitive to minimal pH changes.

**Author Contributions:** Conceptualization, S.C.T., L.F.B., A.R.C.R., R.R.A.S. and T.V.d.O.; investigation, S.C.T.; resources, M.d.P.L. and L.F.B.; data curation, S.C.T.; writing—original draft preparation, S.C.T.; writing—review and editing, L.F.B., A.R.C.R., T.V.d.O. and M.d.P.L.; project administration, S.C.T. and T.V.d.O.; funding acquisition, T.C.B.R., P.C.S. and N.d.F.F.S. All authors have read and agreed to the published version of the manuscript.

**Funding:** This work was supported by the Laboratory of Pigments and Bioactive Compounds, Federal University of Viçosa, Brazil; Department of Food Technology, Federal University of Viçosa, Brazil and National Council for Scientific and Technological Development (CNPq) (Process number 138645/2019-1).

**Institutional Review Board Statement:** Not applicable.

**Acknowledgments:** The authors would like to thank Coordination for the Improvement of Higher Education Personnel (CAPES), CNPq and Research Supporting Foundation of Minas Gerais (FAPEMIG) for financial support.

**Conflicts of Interest:** The authors declare no conflict of interest.

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
