# Peer review of "Anthocyanins of Açaí Applied as a Colorimetric Indicator of Milk Spoilage: A Study Using Agar-Agar and Cellulose Acetate as Solid Support to Be Applied in Packaging"

_2673-4176, doi:10.3390/polysaccharides3040041_

Round 1

Reviewer 1 Report

The manuscript entitled “Anthocyanins of açaí applied as a colorimetric indicator of milk spoilage: a study using agar-agar and cellulose acetate as solid support to be applied in packaging” by Samiris Côcco Teixeira, Taila Veloso de Oliveira, Lais Fernanda Batista, Rafael Resende Assis Silva, Matheus  de Paula Lopes, Alane Rafaela Costa Ribeiro, Thaís Caroline Buttow Rigolon, Paulo César Stringheta, Nilda de Fátima Ferreira Soares describes the extraction of anthocyanins, blending it with natural polymer matrix (cellulose or agar-agar) and application as acidity indicator to monitor the freshness of the milk.

I find the results described in the manuscript promising for the development of “smart packaging” and I recommend the publication of this manuscript after a major revision.

Indeed, the importance of this research is high as the authors address (1) the application of natural polymers to food packaging and (2) the production of packaging material with an indicator which monitors the freshness of the packed food. The first one leads to the broader application of renewable feed-stock polymer for packaging and the second ensures food resource preservation.

In my opinion, the quality of the manuscript can be sufficiently enhanced if the authors address the comments listed below.

(1)    My major comment concerns the toxicity of the anthocyanins indicator and its migration to the food from the packaging material. The authors mention “non-toxic” pigments only once in the introduction and assume that the extract which they are using in non-toxic by default. The authors should provide either experimental evidence or the literature reference for the low toxicity of that particular extract. As a second action point, the authors must ensure that the indicator doesn’t migrate to the food and that no additional taste is introduced to the food by the indicator. For that, they need to perform additional experiments on migration.

(2)    Additional point to address is the thermal stability of the indicator. The authors produced their polymer films by solvent casting technique while in the industry the packaging is typically produced via extrusion-based techniques which require polymer melting. Thus the question arises if the indicator is thermally stable to be blended with the polymer via extrusion. The authors should check the thermal stability of the indicator and report it in the manuscript.

(3)    The authors claim that the indicator is dispersed in the polymer matrix after solvent casting on the base of SEM image analysis. In my opinion, SEM is not a suitable technique here. The irregularities and spherical inclusions which authors observe can also be due to the specific structure of the natural polymer films. Two possibilities exist: the authors can prepare control polymer films without the indicator incorporated and examine their structure by SEM. On the basis of the SEM image differences, they can claim the incorporation of the indicator into the polymer matrix. The second possibility is to use FTIR and detect the spectrum which is specific to the indicator. Ideally, both experiments should be performed.

(4)    Prior to publication the authors should proofread and clean up the text of the manuscript. In some places they use “)” for the citations. (For example, lines 179, 195 and 220)      

Author Response

Response to Reviewer 1

Question (1):  My major comment concerns the toxicity of the anthocyanins indicator and its migration to the food from the packaging material. The authors mention “non-toxic” pigments only once in the introduction and assume that the extract which they are using in non-toxic by default. The authors should provide either experimental evidence or the literature reference for the low toxicity of that particular extract. As a second action point, the authors must ensure that the indicator doesn’t migrate to the food and that no additional taste is introduced to the food by the indicator. For that, they need to perform additional experiments on migration.

Response: Dear reviewer, thank you so much for the corrections! Your arguments are very important to us!

About the first argument about the non-toxicity of anthocyanins: We have added references in the paper that indicate that anthocyanins are indicated for human ingestion and reduce various diseases.

About the second argument: The colorimetric indicator developed is for use in industry or regulatory agencies, to check the pH of milk batches. Therefore, we suggest that the colorimetric indicator be used for quality control, i.e. single use.

Question (2):  Additional point to address is the thermal stability of the indicator. The authors produced their polymer films by solvent casting technique while in the industry the packaging is typically produced via extrusion-based techniques which require polymer melting. Thus the question arises if the indicator is thermally stable to be blended with the polymer via extrusion. The authors should check the thermal stability of the indicator and report it in the manuscript.

Response: This question is important. Our colorimetric indicator must be manufactured using the casting method. In our laboratory we have continuous casting equipment and produce a large amount of material. Certainly if it were produced by industry, we would indicate the use of continuous casting only. For our case, the extrusion method for manufacturing the colorimetric indicator is not indicated, because the polymers used in the work, the processing temperature is very high, and when reached the polymer is completely degraded. For better understanding we have added TGA analysis to the manuscript.

Question (3): The authors claim that the indicator is dispersed in the polymer matrix after solvent casting on the base of SEM image analysis. In my opinion, SEM is not a suitable technique here. The irregularities and spherical inclusions which authors observe can also be due to the specific structure of the natural polymer films. Two possibilities exist: the authors can prepare control polymer films without the indicator incorporated and examine their structure by SEM. On the basis of the SEM image differences, they can claim the incorporation of the indicator into the polymer matrix. The second possibility is to use FTIR and detect the spectrum which is specific to the indicator. Ideally, both experiments should be performed.
Response: We have added the SEM Control image (no added anthocyanins) and the FT-IR analysis that proves the presence of the compound.

(4) Prior to publication the authors should proofread and clean up the text of the manuscript. In some places they use “)” for the citations. (For example, lines 179, 195 and 220) 

Response: We have corrected these errors.

Reviewer 2 Report

The work done by S. Teixeira and cols. Entitled “Anthocyanins of açaí applied as a colorimetric indicator of milk 2 spoilage: a study using agar-agar and cellulose acetate as solid support to be applied in packaging” is an interesting proposal, in which the authors made a basic analysis on the use of anthocyanin coated-matrix, either cellulose or agar, as useful colorimetric method to detect changes in pH in order to monitor spoiling food. Using as a principle, colorimetric changes of indicators, due to modifications in their chemical structure when changes in the food are present. The study design is well implemented and developed. Methods used to test their questions are in accordingly and well developed. However, the overall study presents some cavities which need to be addressed.

1)      After quantification of total anthocyanidins from the Acai, it is unclear which and why such concentration was selected, please clarify.

2)       There is no units in the values of total anthocyanidins, please add

3)      Dots in figure 1B, 2B and 3B shouldn’t be line connected; instead, trend line should be evidenced in the linear regression.

4)      Results section fig. 2 cellulose acetate sensor maximum CCR% is about 17 at pH 4.5 with and minimum CCR% 12.5 at pH 7 with a R2=0.77. How do the authors explain such a low decay in the trend, is this result statistical different?

5)      In comparison to the previous question, figure 3 in the results section, agar-agar sensor with anthocyanidins shown a strong CCR% (~33%) and a huge decay at pH7 (~2.5%) with a R2=0.89. Please clarify this central event of the study in the discussion section.

6)       In results section figure 4 it appears that the real difference is describes in left micrographs (AA, AC), showing the surface of the sensors; please explain what do the particles represent in AC and why AA did not show.

Author Response

Response to Reviewer 2

1)      After quantification of total anthocyanidins from the Acai, it is unclear which and why such concentration was selected, please clarify.

Response: Reviewer 2, thanks for the pertinent corrections! There was a typing error. We made the correction.

2)    There is no units in the values of total anthocyanidins, please add

Response: We made the correction.

Questions 3, 4 and 5:

Response: For better understanding we report the data with the coordinates L*, a*, b* and ΔE. Please check if the interpretation is better now.

6)    In results section figure 4 it appears that the real difference is describes in left micrographs (AA, AC), showing the surface of the sensors; please explain what do the particles represent in AC and why AA did not show.

Response: We have added one more image. Please check if it is ok.

Reviewer 3 Report

This study prepared the colorimetric indicator on the basis of combination of anthocyanins of açaí with the polymeric materials for expediently determining milk spoilage. The method is simple and applicable. However, the methodology present in this manuscript is not novelty. The mechanisms underlying the interaction, colorimetric changes and microstructures was not deeply analyzed.

The description of the extraction method for phenolics is unclear and confusing. The extraction reagent is ethanol or pure water? The obtained frozen substances were used for subsequent extraction?

The total anthocyanin contents should be expressed as mg of anthocyanins per 100 g dry basis acai pulp. Please express all data on a dry matter basis! Additionally, where was the total phenolics content data of the obtained samples?

How to decide and obtain the spoiled milk model samples.

Additionally, authors will require help in language and expression revisions throughout the manuscript. For example

Line 91-92: “per mL of extract obtained (mg EAG/100 g of acai pulp).” This requires some editing for more clarity.

Line 94: “the anthocyanin extract (1%), adjusted to pH 7.0, 6.0------”. Meaning of the concentration not clear. What reagents are used to adjust pH.

Line 162: “76.13±14.77; 46.88±25.98 e 20.,3±6.70”. This requires some editing for more clarity.

Line 202-203 and Figure 2: CCR is expressed in unit of percent or not?

All the CCR curve as a function of pH should be revised as linear fitting.

What is the correct rate of judging other milk spoilage?

Author Response

Response to Reviewer 3

Question 1: Review 3, thank you very much for the corrections! We have changed the writing, please check it.

Question 2: We have made the corrections we report the values on a 100g dry basis. We removed the analysis that determines the total phenolic content.

Question 3: For our real-time spoilage analyses, we use lactic acid to promote pH change and simulate spoiled milk. We made a modification to the writing of the results and reported the values in terms of the color coordinates L*, a* and b*.

Question 4: We made the correction.

Question 5: We made the correction.

Question 6: We made the correction.

Question 7,8 and 9: We modified the data treatment and reported the color coordinates, in terms of L*, a* and b*. Check to see if it is ok.

Round 2

Reviewer 1 Report

After the revision, the authors addressed all my comments. In my opinion, the manuscript can be published. 

Author Response

Review 1, thanks for the previous corrections! 
Thank you for nominating our work for publication!

Reviewer 3 Report

All the writing problems have been well resolved and explained. Moreover, the related discussion was supplemented. However, a few writing errors should be checked. For example:

Line 52: Curtis et al. (2009), they →Curtis et al. [9], please check again, and delete the word they.

Line 57: [10–13].

Line 242: The inner illustration is repeated.

Additionally, the fitted curve should be changed to linear one.

Author Response

Dear Reviewer 3, thank you for the new corrections!
Line 52: Curtis et al. (2009), they →Curtis et al. [9], please check again, and delete the word they.
Response: We made the corrections.
Line 57: [10–13].
Response: We made the corrections.
Line 242: The inner illustration is repeated.
Response: We made the corrections.
Additionally, the fitted curve should be changed to linear one.
Answer: Thanks for the corrections! In our work we chose to present our results in colorimetric terms of L*, a* and b*, with ANOVA (Tukey) statistical test

Round 3

Reviewer 3 Report

After several revision, the manuscript can be accepted.